# Hypophosphatemia after Start of Medical Nutrition Therapy Indicates Early Refeeding Syndrome and Increased Electrolyte Requirements in Critically Ill Patients but Has No Impact on Short-Term Survival

**DOI:** 10.3390/nu16070922

**Published:** 2024-03-22

**Authors:** Mathias Schneeweiss-Gleixner, Patrick Haselwanter, Bruno Schneeweiss, Christian Zauner, Marlene Riedl-Wewalka

**Affiliations:** Department of Medicine III, Clinical Division of Gastroenterology and Hepatology, Intensive Care Unit 13H1, Medical University of Vienna, 1090 Vienna, Austria; patrick.haselwanter@meduniwien.ac.at (P.H.); bruno.schneeweiss@yahoo.de (B.S.); christian.zauner@meduniwien.ac.at (C.Z.); marlene.wewalka@meduniwien.ac.at (M.R.-W.)

**Keywords:** refeeding syndrome, ICU, critical care, nutrition, hypophosphatemia

## Abstract

Refeeding syndrome (RFS) is a potentially life-threatening complication in malnourished (critically ill) patients. The presence of various accepted RFS definitions and the inclusion of heterogeneous patient populations in the literature has led to discrepancies in reported incidence rates in patients requiring treatment at an intensive care unit (ICU). We conducted a prospective observational study from 2010 to 2013 to assess the RFS incidence and clinical characteristics among medical ICU patients at a large tertiary center. RFS was defined as a decrease of more than 0.16 mmol/L serum phosphate to values below 0.65 mmol/L within seven days after the start of medical nutrition therapy or pre-existing serum phosphate levels below 0.65 mmol/L. Overall, 195 medical patients admitted to the ICU were included. RFS was recorded in 92 patients (47.18%). The presence of RFS indicated significantly altered phosphate and potassium levels and was accompanied by significantly more electrolyte substitutions (phosphate, potassium, and magnesium). No differences in fluid balance, energy delivery, and insulin requirements were detected. The presence of RFS had no impact on ICU length of stay and ICU mortality. Screening for RFS using simple diagnostic criteria based on serum phosphate levels identified critically ill patients with an increased demand for electrolyte substitutions. Therefore, stringent monitoring of electrolyte levels is indicated to prevent life-threatening complications.

## 1. Introduction

Malnutrition, characterized by an inadequate intake of energy, proteins, and other nutrients, represents a serious problem associated with sarcopenia, weakness, and organ dysfunction. Thereby, malnutrition is known to result in extended hospital length of stay (LOS), elevated short- and long-term mortality, and increasing costs for the healthcare system [1,2,3]. About 20% to 50% of patients are reported to be malnourished at hospital admission [3,4,5]. In patients admitted to intensive care units (ICUs), the number of patients suffering from malnutrition is reported to be even higher, according to recent data (37.8% to 78.1%) [6]. Therefore, medical nutrition therapy (MNT) is of utmost importance in critically ill patients to prevent the disastrous consequences of malnutrition, starvation, and post-aggression metabolism [7]. However, despite its known positive effects, MNT may also cause serious adverse events in malnourished patients. Refeeding syndrome (RFS) is triggered by the restarting of alimentation in malnourished patients. It is characterized by life-threatening complications (e.g., arrhythmias, cardiopulmonary failure, seizures, and coma) induced by severe electrolyte imbalances, a lack of vitamins, and the retention of water and sodium [8,9]. Therefore, rigorous metabolic monitoring is required when MNT is prescribed in malnourished patients.

However, a uniformly accepted definition of RFS is still lacking. Whereas the most widely used definition by Marik et al. defines RFS solely as decreased serum phosphate levels (total serum phosphate levels <0.65 mmol/L or a >0.16 mmol/L decrease in serum phosphate levels to values below 0.65 mmol/L) after the initiation of MNT [10], organ dysfunction and fluid overload in addition to electrolyte imbalances are essential for diagnosing RFS according to Rio et al.’s criteria [11]. Friedli et al. established a more precise definition characterized by three primary criteria (i.e., serum phosphate levels <0.32 mmol/L, magnesium <0.32 mmol/L, and potassium <2.5 mmol/L) and four minor criteria (i.e., serum phosphate <0.81 mmol/L, magnesium <0.74 mmol/L, potassium <3.6 mmol/L, or peripheral edema). For diagnosing RFS, one major or two minor criteria are required [12]. Yet, the authors demand further investigations to confirm organ manifestation and to exclude other reasons for electrolyte imbalance. The 2020 recommendations published by the American Society for Parenteral and Enteral Nutrition (ASPEN) also suggest characterizing RFS according to electrolyte shifts (phosphate, magnesium, and potassium) within the first five days after the restarting of alimentation. In addition, the authors define three grades of RFS (mild, moderate, and severe) based on the severity of the electrolyte imbalance and the presence of decreased thiamin levels and organ dysfunction [13].

There are conflicting data on the prevalence of RFS in ICUs, mainly caused by heterogeneous definitions and patient collectives (medical vs. surgical patients). Moreover, critically ill patients are exposed to a wide variety of invasive procedures, which have a significant impact on electrolyte and fluid balance. As the diagnosis of RFS is primarily based on electrolyte disbalances, these circumstances might induce a bias when screening for RFS. Due to the aforementioned inconsistencies, the current literature reports an incidence ranging from 17% to 52% [14,15,16].

We therefore conducted this prospective observational study of critically ill medical patients admitted to the tertiary ICU of the Medical University of Vienna between 2010 and 2013 in order to assess the occurrence and management of RFS as well as the short-term outcomes of our patients (ICU survival).

## 2. Methods

### 2.1. Study Design and Cohort

We conducted a prospective observational study of all medical patients admitted to a medical ICU at a large tertiary center in Vienna, Austria (Medical University of Vienna/Vienna General Hospital). All adult patients (18 years and older) requiring intensive care treatment for medical reasons and with an estimated ICU length of stay of at least one week were screened for the development of RFS between January 2010 and February 2013. All postoperative patients and patients with contraindications for nutrition support were excluded from our study in order to analyze a homogenous patient cohort (Figure 1). The observation period was set from ICU admission to ICU discharge or death.

This study was approved by the local Ethics Committee of the Medical University of Vienna (ethics vote number: 913/2009) and performed in accordance with the Declaration of Helsinki’s most recent guidelines. Given the study’s observational design, informed consent statements were not required according to national and institutional regulations. 

### 2.2. Nutrition-Related Data and Refeeding Syndrome

All patients in this study received nutrition support according to the ICU’s standard operating procedures, aiming to commence MNT as soon as possible via nasogastric tube after admission. Enteral nutrition (EN) was the preferred route of administration. Parenteral nutrition (PN) was only implemented in patients with contraindications for EN or insufficient EN. The treating physicians set nutrition goals, estimating daily energy requirements using a simple weight-based equation [7].

RFS was defined as a decrease of more than 0.16 mmol/L in serum phosphate concentrations to values below 0.65 mmol/L within seven days or pre-existing serum phosphate levels below 0.65 mmol/L [10]. In order to investigate the rate of RFS and differences in the clinical management of RFS patients compared to patients without RFS, we assessed serum phosphate and magnesium levels once daily. Potassium levels were measured as part of routine blood gas analysis at least three times daily. Moreover, we documented electrolyte changes, electrolyte substitutions, supplied MNT, insulin requirements, energy delivery, and fluid balance in a daily manner for seven days after the commencement of MNT. 

### 2.3. Data Collection

The Vienna General Hospital uses a digital patient data management system (IntelliSpace Critical Care and Anesthesia, Philips, Amsterdam, The Netherlands), which enables complete documentation of all crucial patient characteristics, including diagnosis, physical examination, vital signs, laboratory tests, as well as therapy and medication. The severity of critical illness was calculated within the first 24 h of ICU admission using the Sequential Organ Failure Assessment Score (SOFA) [17] and the Simplified Acute Physiology Score II (SAPS II) [18]. Data on laboratory results (especially electrolytes), life-sustaining therapies (mechanical ventilation, vasopressor support), electrolyte substitution, vitamin substitution, MNT, and fluid balance were documented over the course of seven days after the commencement of nutrition support.

### 2.4. Statistical Analysis

The primary aim of this prospective observational study was to assess the incidence of RFS in patients requiring ICU admission. Predefined secondary objective parameters included the evaluation of exogenous electrolyte requirements and short-term outcomes (i.e., ICU mortality) in patients with RFS compared to non-RFS patients.

Qualitative parameters are shown as absolute numbers with relative frequencies (%). Quantitative parameters are presented as medians with interquartile ranges (IQRs) for non-normally distributed data or as means ± standard deviation for normally distributed data. Non-normally distributed parameters were assessed through visual inspection of histograms. In the case of normal distribution, metric variables were compared between the groups using a t-test. To identify differences in baseline characteristics, Fisher’s exact tests were used to compare categorical variables. The Mann–Whitney U or Wilcoxon signed-rank tests were used for non-parametric variables, as appropriate. The probability of ICU survival was calculated using Kaplan and Meier’s product limit method. The log-rank test determined differences in ICU survival among our study groups. *p*-values < 0.05 were considered statistically significant. All statistical analyses were performed using GraphPad Prism 8 (GraphPad Software, San Diego, CA, USA) and IBM SPSS Statistics 27 (IBM, New York, NY, USA).

## 3. Results

### 3.1. Basic Characteristics

In total, 195 patients (126 male, 64.6%) were included in this study. The basic characteristics are shown in Table 1. At ICU admission, the median age of the study population was 62 (50–71) years, with a median body mass index (BMI) of 26.3 (23.3–30.3) kg/m^2^. The median SOFA and SAPS II scores were 10 (8–13) and 60.0 (48.8–72.0), respectively. The majority of patients (n = 181; 92.8%) required invasive mechanical ventilation (IMV) with a median length of IMV of 9.0 (5.9–18.0) days. The most common route of feeding was total EN (tEN; 78.5%), followed by supplemental PN (sPN; 16.4%) and total PN (tPN; 4.6%). The median ICU-LOS was 14 (6–22) days. In our study population, we report an ICU mortality of 29.2%.

The reasons for ICU admission are depicted in Table 1. Cardiopulmonary resuscitation (37.4%) was the leading cause of ICU admission, followed by respiratory failure (19.5%) and sepsis (19.5%). The frequency of comorbidities is shown in Table 2.

### 3.2. Incidence of RFS

After starting MNT, 92 patients (47.18%) developed RFS according to our applied RFS criteria. Except for height (175 cm vs. 170 cm; *p* = 0.026), we did not find any statistically significant differences concerning the basic characteristics upon comparing patients with and without RFS (Table 1). As shown in Table 2, RFS was found more often in patients suffering from advanced chronic liver disease (22.8% vs. 9.7%; *p* = 0.018) and in patients with alcohol and/or drug abuse (12% vs. 1%; *p* = 0.002).

### 3.3. The Clinical Significance of RFS

Phosphate, magnesium, and potassium levels after the start of MNT are depicted in Figure 2. Serum phosphate levels were significantly lower in the RFS group. The nadir of serum phosphate levels was detected on day 3. Low serum phosphate levels were already present in 26.1% of patients in the RFS group prior to MNT commencement, which is significantly more often compared to the non-RFS cohort (13.6%; *p* = 0.031). Serum potassium levels were also consistently lower in the RFS cohort, especially on days 3 to 5 after the commencement of MNT. Serum magnesium levels were stable in both cohorts over the course of 7 days, with no significant differences (Figure 2). The prevalence of pre-existing low electrolyte levels before the induction of MNT is shown in Table 3.

As seen in Figure 3A, a significantly higher proportion of patients in the RFS group required electrolyte substitution with phosphate, potassium, and magnesium over the course of 7 days after MNT commencement. The daily phosphate, potassium, and magnesium substitutions are depicted in Figure 3B. In addition, the cumulative substitution of all three electrolytes was significantly higher in the RFS cohort (Figure 3C).

The daily mean energy delivery and insulin requirements over the course of 7 days are shown in Figure 4A. Both did not differ significantly between the RFS and non-RFS groups. We did not observe differences in fluid balance, urine output, or fluid supply between both study groups (Figure 4B). All patients in the RFS group received 300 mg of thiamin per day from the time point RFS was identified according to our routine therapeutic regimen.

Although statistical significance was not reached, there was a trend toward lower ICU mortality (22.8% vs. 35%; *p* = 0.083) in the RFS group (Table 1). As assessed by the Kaplan–Meier survival estimation and the log-rank test, the occurrence of RFS did not affect short-term outcomes (Figure 5).

## 4. Discussion

Although RFS represents a known complication in patients receiving nutritional support, data on its incidence in ICUs are scarce and conflicting [10,12,14,19,20,21,22]. One main issue is the lack of a generally accepted definition for RFS, which consequently impedes the management of RFS patients [14,15,16,23]. Various definitions as well as diagnostic criteria have been proposed for RFS, including laboratory alterations, with low serum phosphate levels as a cardinal finding, and different clinical symptoms. In our prospective observational study, which included 195 critically ill medical patients, we aimed to investigate the incidence of RFS by using the diagnostic criteria proposed by Marik et al. and the respective impact on the therapy of these patients [10]. We detected an RFS incidence of 47.18%. The diagnosis of RFS was accompanied by significantly altered magnesium and potassium levels and, consequently, a significantly higher demand for electrolyte substitutions (phosphate, magnesium, and potassium). There were no differences in fluid balance, energy delivery, or insulin requirement upon comparing patients with and without RFS. The occurrence of RFS had no impact on ICU-LOS and ICU mortality.

The reported incidence of RFS in critically ill patients is inconsistent with published rates ranging from 14.6% to 52% [10,12,14,19,20,21,22]. However, comparing these results is difficult, as most of the conducted studies include heterogeneous patient populations and different RFS definitions. Fuentes et al. analyzed surgical ICU patients only and detected RFS in 39% of patients [24]. In contrast, Coskun et al. exclusively included medical patients and reported a higher RFS incidence of 52.14% [19]. Marik et al. and Olthof et al. included both surgical as well as medical patients in their patient cohorts and reported slightly lower RFS incidences of 34% and 36.8%, respectively [10,20]. In the present study, we found an RFS incidence rate of 47.18%. The distinction between surgical and medical ICU patients has major implications regarding nutritional status and, consequently, RFS occurrence. Malnutrition—the main risk factor for the development of RFS—in critically ill patients is reported to have a prevalence of 37.8% to 78.1% [6]. However, the frequency of malnutrition is significantly lower in surgical patients [4,6]. In their systematic review, Lew et al. reported a rate of malnutrition of 5–20% in cardiac surgery patients compared to 82% in patients with acute kidney injury [6]. These differences in the nutritional status and, therefore, the incomparable risk for RFS in surgical and medical ICU patients might at least partly explain the broad incidence range of RFS in ICUs. 

The lack of a consistent RFS definition represents another major problem in the diagnosis and, consequently, management of RFS [14,15,16]. The easily applicable definition of Marik et al. exclusively focuses on alterations in serum phosphate levels after the start of MNT [10]. However, a definition solely based on serum phosphate levels might induce a bias, especially in critically ill patients, where other conditions like alkalosis, insulin and glucose supply, and renal replacement therapy may also cause hypophosphatemia [16]. The real question is whether diagnostic criteria primarily based on electrolyte alteration do justice to RFS as a disease entity or whether it needs additional parameters. Other definitions require, besides electrolyte imbalances, clinical signs such as tachycardia, tachypnea, peripheral edema, acute circulatory fluid overload, or organ dysfunctions, such as pulmonary edema, respiratory failure, or cardiac failure, for the diagnosis of RFS [11,12]. Reber et al. even suggest a differentiation between imminent (only electrolyte disbalances) and manifest (electrolyte disbalances and clinical symptoms) RFS [23]. The practicability most of these definitions is reduced in the context of ICU patients, as critical illness and various intensive care treatment modalities significantly impact the proposed diagnostic criteria due to disease- and treatment-associated fluid/electrolyte shifts in critically ill patients. The definition of Rio et al. demands signs of severe organ dysfunction in order to diagnose RFS [11]. In ICU patients who are already admitted with single- or multi-organ failure, these criteria do not represent valid parameters for RFS diagnosis. The current ASPEN consensus criteria do not require the presence of organ dysfunction for the diagnosis of RFS, and are also easily applicable and more recent than the definition by Marik et al. [13]. However, according to the ASPEN consensus criteria, the requirement for a diagnosis of RFS is already met when a decrease in only one electrolyte compartment (phosphate, magnesium, and/or potassium) is detected [13]. This approach bears the risk of substantial overdiagnosis of RFS in critically ill patients prone to electrolyte disturbances for various other reasons [25]. Despite its limitations, hypophosphatemia is still considered the hallmark of RFS, and the definition of Marik et al., besides being easily applicable, is well established for RFS screening in the intensive care setting and was, therefore, applied in our observational study. In addition, all mentioned diagnostic criteria for RFS, like the ones from Rio et al. (2013), Friedli et al. (2020), and ASPEN (2020), were established after the patient recruitment was finished for our study [11,12,13].

Interestingly, we showed that RFS diagnosis, according to Marik et al., resulted in significant electrolyte disbalances, especially serum phosphate and potassium levels, after the start of MNT in our ICU patient cohort [10]. As a result, patients in the RFS group required significantly more electrolyte substitutions, particularly phosphate, potassium, and magnesium. These results are in line with previously published data using different RFS definitions [20,22]. Although the diagnostic criteria of Marik et al. might not reflect RFS development to its full pathophysiological extent, they sufficiently identify patients with increased risk for electrolyte disbalances after the start of MNT [10]. Consequently, RFS patients require more stringent monitoring of electrolyte levels in order to prevent them from developing more serious complications. 

There are conflicting data on the potential influence of RFS on outcomes in critically ill patients. In our observational study, we did not detect any differences in outcome parameters like ICU mortality and ICU-LOS in patients with RFS. Our data are in line with the study of Olthof et al., who also did not report significant differences in 6-month mortality in their ICU cohort focusing on surgical patients [20]. In contrast, Coskun et al. reported significantly decreased survival rates in ICU patients suffering from RFS [19]. All the mentioned studies are hardly comparable due to divergent patient populations and RFS definitions. Nevertheless, the increased awareness of RFS by using easily applicable definitions as screening tools might explain why the diagnosis of RFS did not negatively influence the outcome of our study population. Early detection of RFS resulted in immediate therapeutic actions like stringent monitoring, electrolyte and thiamin substitutions, and the adaptation of MNT. In this regard, it is reasonable to state that, by using Marik et al.’s definition, we rather identified patients with a high risk for the development of RFS or beginning RFS.

Friedli et al. reported significantly higher mortality rates in RFS patients after 180 days but no differences in 1-month mortality [12]. In the study of Meira et al., no differences in hospital LOS or mortality were reported [22]. However, the studies by Friedli et al. and Meira et al. focused on non-ICU patients with other primary diseases and therapeutic goals, impairing their comparability [12,22].

The present study has several limitations. First, our prospective study was conducted in a single center, limiting the generalizability of the results as they may have been influenced by local routine management by the treating physicians. In addition, a substantial percentage (>90%) of our patient population was mechanically ventilated, likely due to the fact that only patients with an estimated ICU-LOS of at least 1 week were included in the study. Furthermore, patients were treated at the largest tertiary center in East Austria providing the maximum of critical care possible. However, we provide a comprehensive dataset and a relatively large prospective patient cohort, enabling stable results and conclusions. Second, we report a dataset of patients admitted to the ICU from 2010 to 2013. Although there has been an evolution with regard to ICU-specific therapies since 2013, these developments mainly included adapted treatment strategies rather than new drugs or extracorporeal treatment approaches. Indeed, there have been recent advances in medical nutrition therapy, especially in the treatment of patients with RFS, which were not taken into account in our study [23]. However, the impact of these changes cannot be estimated and requires future prospective studies. Third, we have no information on nutrient intake prior to ICU admission. Although these data would provide valuable additional information about the risk of RFS development in our patient cohort, many screening tools that are successfully used in hospitalized patients perform poorly in the ICU setting [26]. Although developed and validated in the ICU setting, the NUTRIC score (nutrition risk in the critically ill) only aims to identify critically ill patients who benefit from aggressive protein–energy provision during their ICU stay [27]. However, it remains questionable whether the NUTRIC score represents a sufficient screening tool for malnutrition, as traditional risk factors for malnutrition (low BMI, history of weight loss, or reduced oral intake) are not included [27].

## 5. Conclusions

In this prospective observational study of 195 critically ill medical patients, RFS, defined using the easily applicable definition of Marik et al. (refeeding-induced hypophosphatemia), was present in about 50% of patients. These patients also featured more electrolyte disbalances and had a significantly higher demand for electrolyte substitutions. The presence of RFS had no impact on ICU survival probably due to the more stringent monitoring of these patients. In conclusion, critically ill patients are at high risk for the development of RFS. However, all currently accepted RFS definitions demonstrate major limitations in the ICU setting, leaving uncertainties regarding the diagnosis and management of RFS. This issue warrants further, preferably prospective and multi-center, studies.

## Figures and Tables

**Figure 1 nutrients-16-00922-f001:**
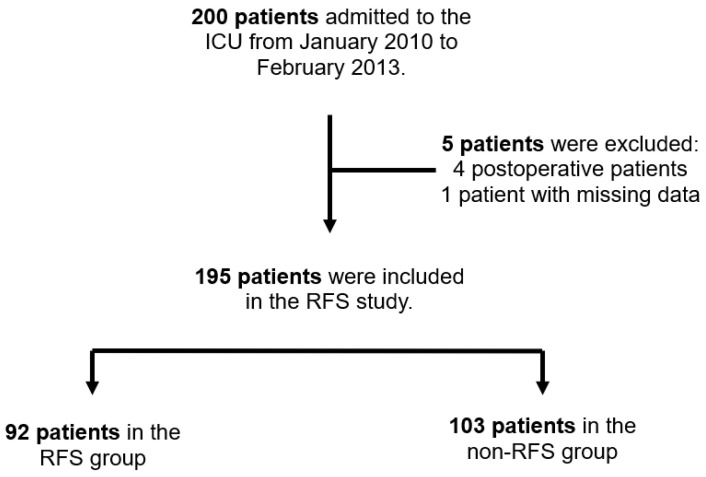
Flowchart of the study population. Abbreviations: ICU, intensive care unit; RFS, refeeding syndrome.

**Figure 2 nutrients-16-00922-f002:**
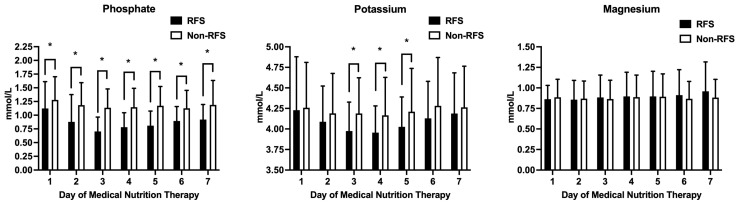
Differences in phosphate, magnesium, and potassium levels in the first seven days after the start of medical nutrition therapy. Mean levels of electrolytes are given in mmol/L. The black columns represent RFS patients, and the white columns represent non-RFS patients. Asterisk (*): *p* < 0.05. Abbreviations: RFS, refeeding syndrome.

**Figure 3 nutrients-16-00922-f003:**
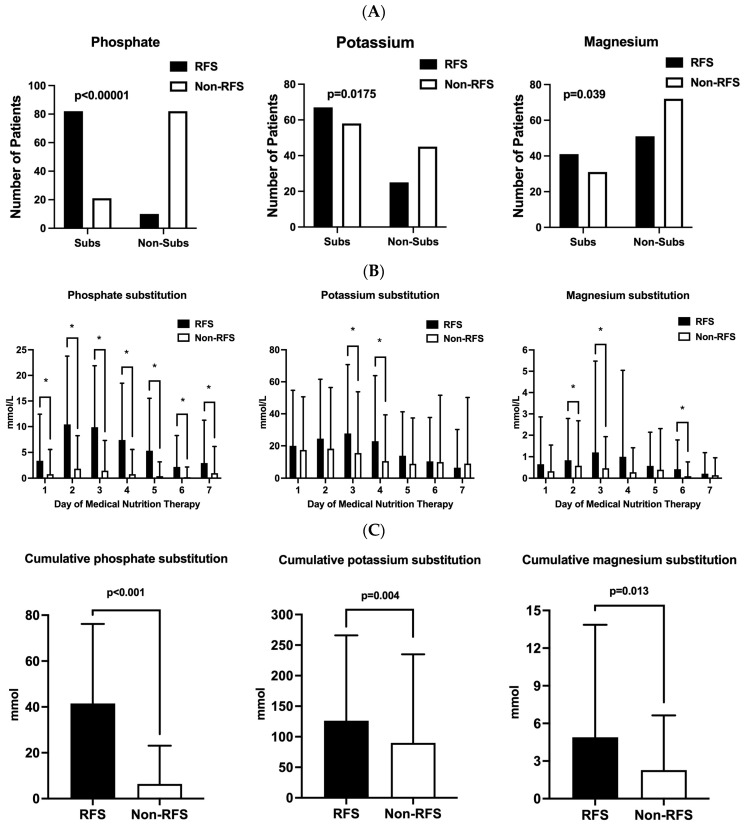
Differences in phosphate, magnesium, and potassium substitutions in the first seven days after the start of medical nutrition therapy and cumulative electrolyte substitutions in patients with RFS (black columns) and without RFS (white columns). Patients requiring electrolyte substitution in the RFS and the non-RFS groups during the observation period are given as an absolute count (**A**). Mean phosphate, magnesium, and potassium substitution levels in the first seven days after starting MNT are given in mmol/L (**B**). Cumulative substitution of electrolytes is given in mmol (**C**). Asterisk (*): *p* < 0.05. Abbreviations: RFS, refeeding syndrome; Subs, substituted patients; non-Subs, patients with no electrolyte substitution.

**Figure 4 nutrients-16-00922-f004:**
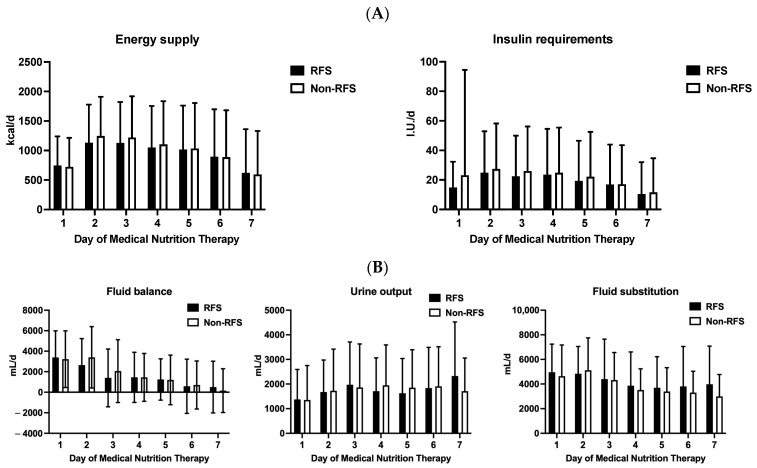
Differences in energy uptake (kcal/d), insulin supply (I.U./d), fluid balance (mL/d), urine output (mL/d), and fluid substitution (mL/d) in the first seven days after the start of medical nutrition therapy in patients with RFS (black columns) and without RFS (white columns). Mean levels of energy delivery and insulin requirements in the first seven days after starting MNT are given in kcal/d and I.U./d (**A**). Fluid balance, urine output, and fluid substitution in the first seven days after starting MNT are given in mL/d (**B**). All parameters did not reach statistical significance. Abbreviations: I.U./d, international units per day; RFS, refeeding syndrome.

**Figure 5 nutrients-16-00922-f005:**
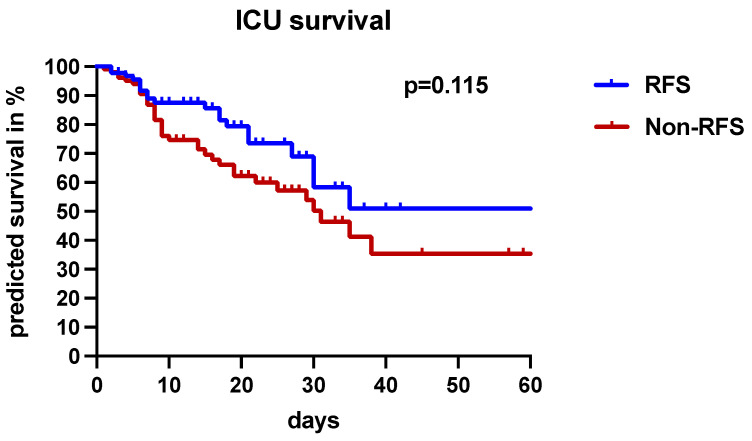
Presence of RFS and ICU survival. The study population was divided into patients with RFS, represented by the blue line, and without RFS, represented by the red line. A Kaplan–Meier-based analysis was used for the calculation of ICU survival. Patients without RFS were associated with a lower predicted survival, but the results did not reach the significance level (*p* = 0.115 according to log-rank test).

**Table 1 nutrients-16-00922-t001:** Basic characteristics among the total study population, RFS group, and non-RFS group.

Basic Characteristics	Total Population (n = 195)	RFS Group (n = 92)	Non-RFS Group (n = 103)	*p*-Value
Age, median (IQR)	62 (50–71)	58 (49–70)	65 (52–73)	0.059
Male, n (%)	126 (64.6)	61 (66.3)	65 (63.1)	0.656
Female, n (%)	69 (35.4)	31 (33.7)	38 (36.9)	0.656
Weight (kg), median (IQR)	80.0 (68.8–94.1)	82 (70–95)	80 (64–93)	0.083
Height (cm), median (IQR)	172.5 (165–180)	175 (165.8–180)	170 (165–179.3)	0.026
BMI (kg/m^2^), median (IQR)	26.3 (23.3–30.3)	26.5 (23.9–30.2)	26.2 (22.0–30.9)	0.325
ICU mortality, n (%)	57 (29.2)	21 (22.8)	36 (35)	0.083
ICU-LOS (days), n (%)	14 (6–22)	15 (6.3–21.8)	11 (6–23)	0.674
SAPS II, median (IQR)	60.0 (48.8–72.0)	60 (51–71)	59.5 (45.0–73.3)	0.688
SOFA, median (IQR)	10 (8–13)	10 (8–12)	10 (7–13)	0.313
IMV, n (%)	181 (92.8)	83 (90.2)	98 (95.2)	0.267
IMV duration, median (IQR)	9 (5.9–18)	18 (4–19)	10 (6–17)	0.190
tEN, n (%)	153 (78.5)	69 (75)	84 (81.6)	0.298
tPN, n (%)	9 (4.6)	6 (6.5)	3 (2.9)	0.311
sPN, n (%)	32 (16.4)	16 (17.4)	16 (15.5)	0.847
Reason for ICU admission				
Cardiopulmonary resuscitation, n (%)	73 (37.4)	31 (33.7)	42 (40.8)	0.374
Cardiovascular event, n (%)	13 (6.7)	4 (4.4)	9 (8.7)	0.261
Respiratory failure, n (%)	38 (19.5)	19 (20.7)	19 (18.5)	0.721
Sepsis, n (%)	38 (19.5)	17 (18.5)	21 (20.4)	0.857
Gastrointestinal bleeding, n (%)	7 (3.6)	4 (4.4)	3 (2.9)	0.709
Neurological failure, n (%)	17 (8.7)	12 (13)	5 (4.9)	0.073
Acute liver failure, n (%)	3 (1.5)	2 (2.2)	1 (1)	0.603
Other, n (%)	6 (3.1)	3 (3.3)	3 (2.9)	>0.999

Abbreviations: BMI, body mass index; ICU, intensive care unit; IMV; invasive mechanical ventilation; IQR, interquartile range; LOS, length of stay; n, number of patients; RFS, refeeding syndrome; SAPS II, Simplified Acute Physiology Score; SOFA, Sequential Organ Failure Assessment Score; sPN, supplemental parenteral nutrition; tEN, total enteral nutrition; tPN, total parenteral nutrition.

**Table 2 nutrients-16-00922-t002:** Comorbidities among the total study population, RFS group, and non-RFS group.

Comorbidities	Total Population (n = 195)	RFS Group (n = 92)	Non-RFS Group (n = 103)	*p*-Value
Cardiovascular disease, n (%)	106 (54.4)	43 (46.7)	63 (61.2)	0.046
DM type I or II, n (%)	28 (14.4)	10 (10.9)	18 (17.5)	0.223
Nicotine abuse, n (%)	10 (5.1)	5 (5.4)	5 (4.9)	>0.999
Alcohol and/or drug abuse, n (%)	12 (6.2)	11 (12)	1 (1)	0.002
Advanced chronic liver disease, n (%)	31 (15.9)	21 (22.8)	10 (9.7)	0.018
COPD, n (%)	26 (13.5)	14 (15.2)	12 (11.7)	0.530
Malignant disease, n (%)	17 (8.7)	5 (5.4)	12 (11.7)	0.137
Neurological disease, n (%)	20 (10.3)	9 (9.8)	11 (10.7)	>0.999
Organ transplantation, n (%)	9 (4.6)	3 (3.3)	6 (5.8)	0.504
Chronic kidney disease, n (%)	14 (7.2)	4 (4.4)	10 (9.7)	0.174
Psychiatric disease, n (%)	3 (1.5)	1 (1.1)	2 (1.9)	>0.999
Immunological disease, n (%)	5 (2.6)	2 (2.2)	3 (2.9)	>0.999
Gastrointestinal disease, n (%)	5 (2.6)	2 (2.2)	3 (2.9)	>0.999

Abbreviations: COPD, chronic obstructive pulmonary disease; DM, diabetes mellitus; n, number of patients; RFS, refeeding syndrome.

**Table 3 nutrients-16-00922-t003:** Absolute and relative frequencies of pre-existing low electrolyte concentrations prior to start of nutrition support in the RFS and non-RFS groups.

Pre-Existing Low Electrolyte Levels Prior to Nutrition Support	RFS Group	Non-RFS Group	*p*-Value
PO_4_ levels ≤ 0.8 mmol/L, n (%)	24 (26.1)	14 (13.6)	0.031
Mg^2+^ levels ≤ 0.65 mmol/L, n (%)	7 (7.6)	9 (8.7)	0.801
K^+^ levels ≤ 3.5 mmol/L, n (%)	9 (9.8)	8 (7.8)	0.312

Electrolyte cut-off values for hypophosphatemia, hypokalemia, and hypomagnesemia were determined based on the reference standards of the local Department of Laboratory Medicine. Abbreviations: K^+^, potassium; Mg^2+^, magnesium; PO_4_, phosphate; RFS, refeeding syndrome.

## Data Availability

The data that support the findings of this study are available from the corresponding authors upon reasonable request due to privacy reasons.

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
