# Peer review of "Hypophosphatemia after Start of Medical Nutrition Therapy Indicates Early Refeeding Syndrome and Increased Electrolyte Requirements in Critically Ill Patients but Has No Impact on Short-Term Survival"

_nutrients, 2024, doi:10.3390/nu16070922_

Round 1
Reviewer 1 Report
Comments and Suggestions for Authors
The study „Hypophosphatemia after start of medical nutrition therapy indicates early refeeding syndrome in critically ill patients and is associated with increased electrolyte requirements but has no impact on short-term survival“ by Schneeweiss-Gleixner and colleagues investigates the very important and not yet fully understood clinical problem of the refeeding syndrome. Authors assessed incidence and clinical characteristics of patients with and without RFS after nutrition therapy admitted to the medical Intensive Care Unit (ICU) of a large tertiary center. The researchers therefore conducted a prospective observational study involving 195 medical ICU patients. Patients who were diagnosed with RFS referring to the applied criteria required more electrolyte substitutions (phosphate, potassium, and magnesium) during their ICU stay, but RFS did not impact ICU length of stay or ICU mortality.
The paper is very well written and explains the conducted study in a clear and well understandable manner. However, several issues need to be discussed:
Major remarks:
- Authors completely correctly state that “Although RFS represents a known complication in patients receiving nutritional support, data on its incidence at the ICU is scarce and conflicting. One main issue is the lack of a generally accepted definition for RFS, which consequently impedes the management of RFS patients.“ But in their own study they are using the RFS definition of the 1996 publication by Marik et al. (“RFS was defined by a decrease of more than 0.16 mmol/L of serum phosphate concentrations to values below 0.65 mmol/L within seven days or preexisting serum phosphate levels below 0.65 mmol/L.“), and not the present ASPEN consensus definition (p. 3, ll. 103-106). The discussion in paragraph 3 of the discussion-part to me is not convincing, as the ASPEN consensus criteria (“A decrease in any 1, 2, or 3 of serum phosphorus, potassium, and/or magnesium levels by 10%–20%(mild RS), 20%–30% (moderate RS), or>30% and/or organ dysfunction resulting from a decrease in any of these and/or due to thiamin deficiency (severe RS). And occurring within 5 days of reinitiating or substantially increasing energy provision) can be met without including organ function, just by interpreting electrolytes and onset of electrolyte imbalances.
- Furthermore, in their cohort I understood that electrolyte imbalances were compensated by electrolyte infusion once detected. Thus I would interpret their data not as patients with and without RFS who do not show differentiated outcomes (Fig. 5), but patients without RFS vs. patients with high risk of or beginning RFS but who were diagnosed and treated very early and thus did not have a worse outcome then the control group.
- Comparing the baseline characteristics of the assessed cohort of ICU admitted patients, the number of patients receiving invasive mechanical ventilation of >90% seems to be very high compared to other observations (e.g., around 50 % in https://pubmed.ncbi.nlm.nih.gov/30339549/). Authors should discuss that.
- For developing a RFS, patients need to have a micronutrient deficiency, usually due to chronic malnutrition, recent lack of nutrient consumption, or both, before nutrition is reintroduced. In their study, authors do not provide information about nutrient intake prior to ICU admittance. If not available, they should discuss that as a major limitation for distinguishing the observed electrolyte imbalances due to RFS from other conditions occurring at an ICU (possible causes are already explained in the manuscript)
- It should be stated how often the electrolytes used for the definition of RFS were measured. Was this standardized for all patients, or was there a difference, maybe explaining different detection rates of RFS?
Minor remarks:
- The title is lengthy and could be shortened
- P.1, l. 42: there should not be a space between digits and the %-symbol
- Sometimes the unit is written only once when intervals are given, sometimes twice (e.g., p.1, l. 42 and pp. 1/2, ll. 44/45). This should be standardized.
- When speaking of electrolyte concentrations (e.g., p. 2, l. 56) authors should use the term “concentration” rather than “level”
- Legend Fig. 1: “syndrome” should be lower case; “MNT, medical nutrition therapy” is not needed. Abbreviations should be in alphabetical order.
- Authors should explain why posttraumatic patients were excluded from the study (p. 2, l. 87). I understand why postoperative patients, who probably often were inpatient before ICU admittance were excluded. However, if I read Fig. 1 correctly, there were no posttraumatic patients anyway
- Authors should report if the study was preregistered (if not done, at least to the editor if not yet done)
- Table 1 legend: abbreviations should be in alphabetical order.
- Table 2: the sums for “total population” of patients with type 1 and 2 diabetes and with COPD do not seem to be correct
- Table 2 legend: abbreviations should be in alphabetical order.
- Figures 2, 3A,B, and 4B should be depicted in a larger scale to be well legible.
Comments on the Quality of English Language
See minor comments above
Reviewer 2 Report
Comments and Suggestions for Authors
The manuscript by Mathias et al presents a prospective observational study investigating the incidence of refeeding syndrome (RFS) in critically ill patients in an ICU setting, focusing on hypophosphatemia as an early indicator. The study found a significant incidence of RFS, characterized by altered electrolyte levels and increased electrolyte substitution requirements, though it did not impact short-term ICU survival.
The study addresses a critical and underexplored area in critical care nutrition and provides valuable insights into the management of RFS. The authors employ a prospective observational design and clear, operational definitions for RFS based on serum phosphate levels. Additionally, detailed statistical analysis offers a thorough examination of the relationship between RFS and electrolyte management.
However, I have some reservations that need to be clarified. After presenting several references for varying criteria to diagnose RFS, the authors have failed to inform the reader why they picked one criteria over the others while conducting this prospective study. The study’s focus on hypophosphatemia to define RFS, could potentially limit the understanding of RFS's full clinical spectrum, whereas incorporating broader diagnostic criteria could provide a more comprehensive understanding of RFS.
While the study effectively addresses short-term outcomes, an exploration of long-term survival and recovery post-ICU would have added more depth to the findings. If the authors wish to restrict themselves to the short-term outcomes only, they should mention so in their objectives.
Another opportunity for improvement would be to compare their management with other nutritional management strategies or a control group not using the specific RFS criteria. This could offer insights into the effectiveness of different approaches. Again, if the authors do not wish to do so, they need to be more explicit in their objectives. Their objectives are very open ended and non-specific at present.
Conducted in a single center almost a decade ago, the findings may not be broadly applicable at this time since nutritional therapies have become more aggressive and nuanced with passing time.
Comments on the Quality of English Languageacceptable
Round 2
Reviewer 1 Report
Comments and Suggestions for Authors
With their “Response to Reviewer #1”-documant authors have very comprehensively responded point-by-point to all raised issues. However, some minor remarks still remain:
All in the review addressed limitations have been acknowledged by authors and respective paragraphs were added to the discussion. However, the abstract, introduction and conclusion have not been edited accordingly:
o In my opinion it needs to be additionally clarified at least also in the conclusions that not the ASPEN consensus definition, but the older definition by Marik et al. was used. Also, the sentences “The lack of a uniformly accepted RFS definition and heterogeneous patient populations in the literature has led to discrepancies in reported incidence rates in patients requiring treatment at an intensive care unit (ICU)” in the abstract and „The lack of a uniformly accepted RFS definition makes diagnosis and management difficult, warranting further, preferably prospective and multi-center studies.“ in the conclusion should be changed, as authors themselves did not use the ASPEN consensus definition, and therefore in my opinion the call for a consented RFS definition here seems misplaced.
o Another major limitation of a paper published in 2024 is that data is from 2010-2013, with according diagnostic and therapeutic state of the art of a decade ago, as authors correctly state in their discussion, was used. However, neither the abstract nor the introduction mention this issue. On the contrary, when reading “We conducted a prospective observational study..” in the abstract this to me suggests a recent study. In my opinion, both abstract and introduction should make clear that analyzed data is from 2010-2013.
Reviewer 2 Report
Comments and Suggestions for Authors
Mathias et al have responded to all my concerns and made appropriate changes. I agree with the changes they have made to their objectives to be more specific and in-sync with their actual methodology. I also agree with their text modifications that provide justification for utilizing the older RFS diagnostic criterion.
